# Preparation and Properties of Electrospun PLLA/PTMC Scaffolds

**DOI:** 10.3390/polym14204406

**Published:** 2022-10-18

**Authors:** Dengbang Jiang, Haoying Zou, Heng Zhang, Wan Zhao, Yaozhong Lan, Mingwei Yuan

**Affiliations:** National and Local Joint Engineering Research Center for Green Preparation Technology of Biobased Materials, Yunnan Minzu University, Kunming 650500, China

**Keywords:** L-polylactic acid, poly(trimethylene carbonate), electrospun fibres, tissue engineering

## Abstract

Poly(L-lactide) (PLLA) and PLLA/poly(trimethylene carbonate) (PTMC) scaffolds characterised by different PLLA:PTMC mass ratios (10:0, 9:1, 8:2, 7:3, 6:4 and 5:5) were prepared via electrospinning. The results showed that increasing the PTMC content in the spinning solution caused the following effects: (1) the diameter of the prepared PLLA/PTMC electrospun fibres gradually increased from 188.12 ± 48.87 nm (10:0) to 584.01 ± 60.68 nm (5:5), (2) electrospun fibres with uniform diameters and no beads could be prepared at the PTMC contents of >30%, (3) the elastic modulus of the fibre initially increased and then decreased, reaching a maximum value of 74.49 ± 8.22 Mpa (5:5) and (4) the elongation at the breaking point of the fibres increased gradually from 24.71% to 344.85%. Compared with the PLLA electrospun fibrous membrane, the prepared PLLA/PTMC electrospun fibrous membrane exhibited considerably improved mechanical properties while maintaining good histocompatibility.

## 1. Introduction

Since the beginning of the 21st century, oil resources have become increasingly scarce, and the extensive use of petroleum-based plastics has caused a growing problem of serious environmental pollution; thus, researchers have been focussing on the development of biodegradable bio-based plastics. As a new type of green plastic, poly(L-lactide) (PLLA) exhibits good biodegradability and biocompatibility. PLLA can be polymerised from molecular lactic acid fermented by biomass raw materials. Given its renewability, high strength, high modulus, thermoplastic and easy processability, PLLA is the most widely studied and applied thermoplastic polyester in the biomedical field [1,2,3].

Electrospinning technology is a direct, widely used and relatively easily applied method for the preparation of nanofibers [4,5,6,7,8,9]. Notably, electrospun fibre membranes exhibit high surface area ratio and porosity. These membranes can simulate the natural extracellular matrix (ECM), thus promoting cell adhesion, migration and proliferation [10,11]. In recent years, electrospun PLLA fibre membranes have been actively developed and applied in biomedical fields; for instance, in the manufacture of tissue engineering scaffolds and as drug delivery systems [12,13]. Shamaz et al. [14] developed a novel PLLA-3D scaffold that mimics bone in chemical composition and fibrous architecture. Monticelli et al. [15] dispersed synthetic talc into a PLLA electrospinning fibre to obtain a PLLA fibre characterised by a superhydrophobic surface. Zhou et al. [16] implemented the electrostatic layer-by-layer self-assembly method to deposit chitosan, heparin and graphene oxide on an electrospun PLLA nanofiber membrane to modify it. Shankar et al. [17] prepared a curcumin-containing PLLA fibre membrane as a potential wound-dressing material via electrospinning. Although the bioactivity of the PLLA fibre membrane has been modified a number of times, this material’s poor toughness, lack of flexibility and elasticity and insufficient mechanical properties limit the applications of PLLA [18].

The mechanical properties of PLLA are known to improve because of chemical crosslinking, plasticisation and blending. Indeed, blending has become the main method utilised to improve the mechanical properties of PLLA, because of its simplicity, economy and effectiveness. For example, Qi et al. [19] prepared HAP (hydroxyapatite)-containing PLLA/PCL (polycaprolactone) nanofibre scaffolds via electrospinning for in vivo conditions to simulate the natural bone extracellular matrix; the goal of this approach was to develop materials with potential applications as bone tissue engineering scaffolds. In fact, the addition of PCL was observed to increase the toughness of the PLLA fibre membrane. López et al. [20] improved the properties of PLLA by blending it with thermoplastic starch. Nazir et al. [21] improved the properties of PLLA by blending it with a cyclo-olefinic copolymer. Jiang et al. [22] prepared PMMA branched PCL polymer and modified the PLLA fibre membrane with it.

Poly(trimethylene carbonate) (PTMC) is a newly developed biodegradable material belonging to the linear aliphatic polyester family of compounds, and it is similar to PLLA. However, PTMC has a superior elasticity than PLLA; it also exhibits a high level of toughness at room temperature and for in vivo conditions. Moreover, no acidic intermediates are formed in the process of PTMC degradation, so it will not automatically accelerate degradation [23]. Several countries allow the use of PTMC in the medical field (including the FDA certification in the United States). This material has been used for nerve repairing [24], sustained drug release [25], the manufacture of three-dimensional tissue engineering scaffolds and in other medically relevant applications [26,27,28].

Materials that exhibit both strength and flexibility can be obtained by blending PTMC and PLLA homopolymers or via co-polymerisation of their monomers [5,6]. Each method has its own advantages. Blends are straightforward to prepare, for example, by dissolution in common solvents or co-extrusion, and the proportion of respective polymers can be easily controlled. Materials based on copolymers are more homogenous than blends; however, their synthesis can be complex. Previous literature has yet to report efforts toward improving the tenacity of PLLA electrospinning fibres by adding PTMC to PLLA.

Therefore, in this study, a PLLA/PTMC electrospun fibre membrane material was prepared by physical blending of PTMC and PLLA utilising the electrospinning technology. Scanning electron microscopy, thermal analysis, uniaxial tensile testing, static water contact angle determination and MTT (3-(4,5-dimethylthiazol-2-yl)-2,5-diphenyltetrazolium bromide) assay were employed to establish the morphological characteristics, thermal stability, mechanical properties, wettability and cytotoxicity of the prepared membrane materials. Our main purpose herein is to provide basic data for expanding the scope of application of degradable PLLA/PTMC materials in biomedical fields, such as bio-absorbable sutures, implantable medical devices, tissue engineering scaffolds and controlled drug delivery systems.

## 2. Materials and Methods

### 2.1. Materials

PLLA (Mn = 111,514) was purchased from Anhui Fengyuan biochemistry Co., Ltd. (Bangbu, China), and PTMC (Mn = 231,068) was self-made by the laboratory. Dichloromethane (DCM) of analytical purity was purchased from Tianjin Damao Chemical Reagent Co., Ltd. (Tianjin, China); *N*,*N*-dimethylformamide (DMF) of analytical purity was purchased from Tianjin kemiou Chemical Reagent Co., Ltd. (Tianjin, China).

The preparation and purification of PTMC was conducted as follows. An amount of 10 g of TMC monomer and 0.1 g of Sn(Oct)_2_ in toluene were added to a dry glass reaction flask. Toluene was removed by bubbling dry nitrogen through the reaction flask and then evacuating the flask (repeated 3 times), and then sealing the flask under vacuum conditions. The reaction flask was placed in a constant-temperature oil bath at (130 ± 2) °C for 24 h, after which the flask was allowed to cool naturally at 25 °C. Then, the reaction flask was opened to allow sampling of the polymer, which was dissolved in 50 mL of chloroform. The resulting solution was poured into 200 mL of methanol for precipitation. The precipitate was filtered and dried under vacuum at 37 °C to a constant mass to afford the product PTMC.

### 2.2. Preparation of PLLA/PTMC Fibres by Electrospinning

A series of 4-wt% polymer solutions were prepared in a mixed solvent comprising DCM and DMF (mass ratio 3:1, respectively) as solvent and PLLA and PTMC as solutes. The prepared products were divided into six groups according to the mass ratio of PLLA and PTMC: PLLA material group, 9:1 PLLA/PTMC material group (PLLA and PTMC mass ratio 9:1), 8:2 PLLA/PTMC material group (PLLA and PTMC mass ratio 8:2), 7:3 PLLA/PTMC material group (PLLA and PTMC mass ratio 7:3), 6:4 PLLA/PTMC material group (PLLA and PTMC mass ratio 6:4) and 5:5 PLLA/PTMC material group (PLLA and PTMC mass ratio 5:5). The various polymer solutions thus obtained were thoroughly stirred, and each of them was loaded into a 5 mL syringe with a 20 g stainless steel needle; the fibre membrane material was then prepared using an electrostatic spinning machine (YFSP-T, Tianjin Yunfan Technology Co., Ltd., Tianjin, China). The applied spinning parameters were the following: receiving distance, 10 cm; pushing speed, 0.0030 mm/s; output end, 10 kV; receiving end, 2 kV; duration, 12 h. The prepared film materials were vacuum-dried for 24 h to remove residual organic solvents and moisture and then sealed and packed for later use [19].

### 2.3. Surface Morphology Analysis of PLLA/PTMC Electrospun Fibers

After vacuum-drying and gold-spraying, the surface morphological characteristics of PLLA/PTMC electrospun film materials were determined using a scanning electron microscope (NOVA NANOSEM-450, FEI company, Hillsboro, OR, USA).

### 2.4. Thermal Analysis of PLLA/PTMC Electrospun Fibres

Thermogravimetric analysis (TGA) was conducted on the materials using a thermal analyser (NETZSCH STA2500). In the TGA experiments, the sample mass was ~7 mg. The heating rate was 10 °C/min. The applied atmosphere comprised nitrogen. The gas flow rate was 100 mL/min, and the test temperature range was 30–600 °C. Differential scanning calorimetry (DSC) experiments were conducted using a DSC 214 polymer instrument supplied by the NETZSCH Group. In these experiments, the sample volume was ~7 mg, the heating rate was 10 °C/min, the applied atmosphere consisted of nitrogen, the flow rate was 100 mL/min and the test temperature range was 30–500 °C.

### 2.5. Water Contact Angle Test of PLLA/PTMC Electrospun Fibres

At 25 °C, the water contact angle of the fibre membrane was measured using the JY-PHa video optical contact angle tester (Chengde Youte Testing Instrument Manufacturing Co., Ltd., Chengde, China) to quantify the wettability of the fibre membrane. In detail, the newly prepared fibre film was dried under vacuum for 24 h. It was then cut into a standard sample film 1 cm × 1 cm in size, paved and placed on the sample plate. Afterwards, a drop of double-distilled water (5 μL) was placed on the surface of the sample, and three positions were selected on the surface of each sample film to measure the contact angle. The average value was determined and the standard deviation was calculated.

### 2.6. Mechanical Property Test of PLLA/PTMC Electrospun Fibre Membrane Materials

The uniaxial tensile test was conducted on PLLA/PTMC and PLLA electrospun film materials by using the universal material testing machine (CMT6104, American MTS Co., Ltd., Monroe, NC, USA). Briefly, the sample was fixed on the standard tablet press. The initial tensile length was set to 20 mm and the tensile speed was set to 2 mm/min; the material stress–strain curves were then drawn; the fracture strain was subsequently recorded and calculated, and so were tensile stress, elastic modulus and other parameters.

### 2.7. Cytotoxicity Tests Conducted on PLLA/PTMC Electrospun Fibres

The cytotoxicity of the material was assessed by the MTT assay. The prepared fibre membranes were cut into 8 mm × 8 mm membranes and sterilised by ultraviolet light for 2 h for use. In a 96-well cell culture plate, MC3T3-E1 cells (Saibokang Biological (China)) were seeded at 1 × 104 cells/well in α-MEM medium containing 10% foetal bovine serum (BOVOGEN (South America)). Cells were incubated for 24 h at 37 °C and 5% CO_2_. Then, sterilised fibre membrane material was added to each well, and the incubation was continued for 72 h. The original medium was aspirated, 200 µL of medium containing 0.5 mg/mL MTT was added to each well and it continued to culture for 4 h. The medium and MTT solution were removed, 150 µL dimethyl sulfoxide (DMSO) was added, shaken for 10 min and then absorbance was measured at 570 nm with a microplate reader. The cell relative growth rate (RGR) was calculated using the following equation: RGR (%) = ODE/ODC × 100%, where ODE is the absorbance of the prepared sample group and ODC is the absorbance of the control group. According to the standard of ISO 10993-5, a value for RGR below 75% is indicative of a degree of toxicity for the cells [29]. We performed three sets of parallel experiments to calculate the mean standard deviation.

### 2.8. Biodegradation Analysis

The PLLA/PTMC electrospinning scaffold material was immersed in a phosphate buffer solution (PBS; pH = 7.4) at 37 °C and its biodegradation performance was evaluated by measuring the remaining mass of the material after 1, 3, 7, 14 and 21 days. The degradation experiments were repeated 5 times to ensure accuracy.

All data in figures were expressed as mean ± standard deviation (SD). Statistical Product Service Solutions software (SPSS) was used for statistical analysis of the relevant data.

## 3. Results and Discussion

### 3.1. Morphological Characteristics of PLLA/PTMC Electrospun Fibres

Surface morphology is one of the most important characteristics of electrospun materials. Figure 1 presents the SEM image of the fibres. Figure 2 shows the average diameter of the fibres prepared with different blend compositions [30]. The results of scanning electron microscopy showed that the diameter and uniformity of fibre samples prepared with different mass ratios of PLLA and PTMC were quite different. The average diameter of the PLLA electrospun fibre membrane was 188.12 ± 48.87 nm (see Figure 1A). The surface morphologies of PLLA/PTMC (9:1) and PLLA/PTMC (8:2) groups were similar to that of the PLLA group (see Figure 1B,C). The average diameters of the electrospun fibre membranes of the PLLA/PTMC (9:1) and PLLA/PTMC (8:2) groups had values of 333.38 ± 78.49 nm and 393.92 ± 75.22 nm, respectively. The main reason is that the content of PTMC in the spinning solution is less, which makes the viscosity of the spinning solution low, and the spinning jet is unstable, resulting in insufficient solvent volatilisation and adhesion between fibres. However, as the PTMC content of the material increased, the viscosity of the spinning solution gradually increased, the fibre uniformity increased and the adhesion decreased. In the case of the PLLA/PTMC (7:3) group, the adhesion disappeared completely (see Figure 1D). The surface morphologies of the PLLA/PTMC (6:4) (see Figure 1E), PLLA/PTMC (5:5) (see Figure 1F) and PLLA/PTMC (7:3) groups were similar to each other (see Figure 1E,F). As the PTMC content increased, the fibre diameters and the sizes of the pores between fibres gradually decreased, while the level of uniformity in diameter distribution increased. The mean fibre diameters of PLLA/PTMC (6:4), PLLA/PTMC (5:5) and PLLA/PTMC (7:3) were (453.26 ± 70.31) nm, (513.88 ± 66.96) nm and (584.01 ± 60.68) nm, respectively.

### 3.2. Thermodynamic Properties of PLLA/PTMC Electrospun Fibres

To better understand the compatibility between PTMC and PLLA, their aggregation structure in fibre membranes was measured by DSC. The crystallinity (Xc) of PLLA can be calculated according to the following equation: Xc = (ΔHm/93.7 J/g) × 100%, where ΔHm represents the melting enthalpy (J/g) and 93.7 J/g is the melting enthalpy of 100% crystalline PLLA [31]. The degree of crystallinity was recalculated for the actual amount of the PLLA in the sample because only PLLA undergoes crystallisation.

Figure 3 presents the DSC curves obtained for PLLA/PTMC fibre membranes characterised via different blending ratios and the Xc values calculated for the different PLLA/PTMC fibre membrane samples. As shown by the data reported in Figure 3B, as the PLLA/PTMC mass ratio in the fibre membrane samples increased, the crystallinity of PLLA decreased monotonically from 50.35% to 46.01–25.74%, which means that the crystallisation of PLLA may be affected by PTMC. In fact, PTMC prevents the neat folding and rearrangement of PLLA molecular chains, resulting in a reduction of the tendency of PLLA to crystallise [32].

The results of TGA experiments indicate that PLLA weight loss mainly occurred in the 325–395 °C temperature range, and the weight loss 50% temperature (TD50%) was 370.53 °C; the weight loss of PTMC occurred mainly in the 245–300 °C temperature range, and the TD50% was 280.57 °C; the temperature range for the weight loss of the PLLA/PTMC composite scaffold was observed to be located between that of PLLA and that of PTMC. The TD50% value for PLLA/PTMC (9:1) was 367.67 °C, that for PLLA/PTMC (8:2) was 355.38 °C, that for PLLA/PTMC (7:3) was 343.55 °C, that for PLLA/PTMC (6:4) was 343.27 °C and that for PLLA/PTMC (5:5) was 336.79 °C (see Figure 4).

The results of the DSC and TGA experiments indicate that the addition of PTMC to PLLA reduces the heat resistance of the obtained fibre membrane. Moreover, although as the PTMC content in the blend film increases, the heat resistance of the composite decreases, the described change has little effect on the application of the composite as medical stent material.

### 3.3. Mechanical Uniaxial Tensile Tests Conducted on PLLA/PTMC Electrospun Fibre Membrane Materials

PLLA electrospun fibre membranes can meet the needs of a broad range of biomedical applications only if they maintain a sufficient degree of mechanical stability. Therefore, the mechanical properties of the obtained PLLA/PTMC electrospun fibre membranes were studied [33]. In Figure 5, data reflecting the effect that the PLLA/PTMC blending ratio had on the tensile strength and elongation at the break of the various film materials are reported. As the PTMC content in the said materials increased, their tensile strength and elongation at the break increased. The breaking strain and tensile stress of the fibre membrane gradually increased from 24.71% ± 0.76% and 0.49 ± 0.07 MPa in PLLA to 344.85% ± 32.70% and 8.81 ± 1.40 MPa in PLLA/PTMC (5:5). As can be evinced from the data reported in Table 1, as compared with the PLLA fibre membrane scaffold material, the corresponding fibre membrane materials containing PTMC exhibited improved mechanical properties and substantially increased toughness. With an increase in the PTMC content of the hybrid fibre membrane scaffold material, its toughness increased.

The elastic modulus of the spun fibre membrane material first increased with the increasing PTMC content and then suddenly decreased when the PTMC content reached 50%. For PLLA and PLLA/PTMC (9:1), although their crystallinity is higher, the prepared fibres have small diameters, poor uniformity and large inter-fibre pore spaces, resulting in a low fibre elastic modulus. For 8:2 PLLA/PTMC–6:4 PLLA/PTMC, although the crystallinity of the fibres gradually decreases, their diameter increases gradually; they exhibit good uniformity and high density, and the elastic modulus of their membrane gradually increases. In the case of the PLLA/PTMC (5:5) fibre membrane, when the PTMC content reached 50%, the crystallinity of PLLA in the fibres decreased to 25.7% and the fibre membrane exhibited a high elasticity and low modulus similar to PTMC. The modulus also suddenly decreased.

### 3.4. Wettability Experiments Conducted on PLLA/PTMC Electrospun Fibre Membrane Materials

The results of wettability and water contact angle experiments conducted on different fibre membranes are reported in Figure 6 (the tangent angle measurement method was applied) [34,35]. The water contact angles of the PLLA and PLLA/PTMC fibre membranes characterised via different blending proportions of the two components have values greater than 90°. The water droplets observed on the surface of several fibre membranes were round, indicating that the pure PLLA fibre membrane and the two-component PLLA/PTMC fibre membranes are hydrophobic. Notably, the PTMC content of the various fibre membranes had no significant effect on the water contact angle of the said samples. The water contact angle of the pure PLLA fibre membrane was 128.0°, a rather similar value to those measured for the PLLA/PTMC (9:1) and PLLA/PTMC (8:2) fibre membranes (127.2° and 127.5°, respectively). The water contact angles of PLLA/PTMC (7:3), PLLA/PTMC (6:4), PLLA/PTMC (5:5) and PTMC fibre membranes were quite similar to each other, at measured values of 122.1°, 121.1°, 121.2° and 120.8° respectively. Therefore, the addition of PTMC to the fibre membranes can be concluded to have little effect on the wettability of the PLLA fibre membrane. The small changes in wettability measured among the first three fibre membranes mentioned above and the last three fibre membranes may be due to differences in surface morphology between the fibre membranes.

### 3.5. Cytotoxicity of PLLA/PTMC Electrospun Fibre Membrane Materials

As potential scaffold materials for tissue engineering, the prepared fibrous membranes should have good cell compatibility. MTT assays were thus conducted to determine the cytotoxicity of the prepared fibre membranes. Taking the cell culture medium as the negative control, the RGRs of the cell cultures containing PLLA and PLLA/PTMC samples characterised via different blending ratios were measured (see Table 2 and Figure 7) [36]. MTT assays indicate that the electrospun PLLA/PTMC scaffolds induced insignificant cytotoxic effect. MC3T3-E1 cells were cultured on PLLA/PTMC scaffolds for comparison with PLLA scaffolds. Cells plated on PLLA/PTMC scaffolds showed a proliferation rate comparable to that of normalised controls and higher than that of PLLA scaffolds on day 3.

To further evaluate the cytotoxicity of the electrospun PLLA/PTMC scaffolds, live and dead cells after three days of culture were double-stained with Calcein-AM and PI and visualised using confocal microscopy. Figure 8 shows fluorescent photographs of dead and living cells captured via a laser confocal microscope. In the figure, live and dead cells present green (strongly) and red fluorescence, respectively. Figure 8 shows that the cells are well distributed on the surface of the PLLA/PTMC spun fibre membrane and the number of dead cells is very low. Meanwhile, the live/dead cell images of cells cultured on PLLA/PTMC spun fibre membranes with different mass ratios for three days showed similar branched and interconnected network structures, indicating an increased cell proliferation rate [37]. This further proves that the PLLA/PTMC electrospun scaffolds have good cytocompatibility under different mass ratio compositions.

Overall, the PLLA fibre membrane and the PLLA/PTMC blend fibre membranes have good biocompatibility, so they can be used as scaffold materials for tissue engineering.

### 3.6. Biodegradability

All samples exhibited a linear increase in weight loss with increasing soaking time in PBS (Figure 9). After 21 days of degradation, the observed weight losses of PLLA/PTMC spun fibres at the mass ratios of 10:0, 9:1, 8:2, 7:3, 6:4 and 5:5 were 31%, 35%, 38%, 42%, 44% and 48%, respectively. With increasing PTMC content, the degradation rate of the material increased gradually due to the decrease in the crystallinity of PLLA and a slight increase in the hydrophilicity of the material owing to the increase in the PTMC content, causing the material to become more vulnerable to attack by water molecules in the PBS solution. Therefore, the prepared PLLA/PTMC electrospun fibre membranes can be regarded as biodegradable materials. The introduction of PTMC accelerated the degradation rate of the material, while the degradation rate of the material could be adjusted by controlling the amount of PTMC added.

## 4. Conclusions

In this study, PLLA/PTMC blend fibre membranes designed to be used for tissue engineering were prepared via electrospinning for the first time. The morphology, thermodynamic characteristics, mechanical properties, wettability, biocompatibility and biodegradability of the obtained blends were investigated in detail. The results of the investigations indicate that the diameter and uniformity of spun fibres increased with increasing PTMC content. The addition of PTMC resulted in a considerable increase in the toughness of the PLLA fibrous membrane, while the wettability of the blend films did not change substantially with the addition of PTMC to the blends. At the same time, the PLLA/PTMC blend fibre membranes exhibited no cytotoxicity and adjustable biodegradation rate; therefore, they are potentially superior choices to the pure PLLA fibre membrane for use in tissue engineering. The PLLA/PTMC electrospun fibre scaffold containing 40% PTMC has high elastic modulus and excellent toughness and is expected to be applied in bone tissue engineering scaffolds and cell culture scaffolds.

## Figures and Tables

**Figure 1 polymers-14-04406-f001:**
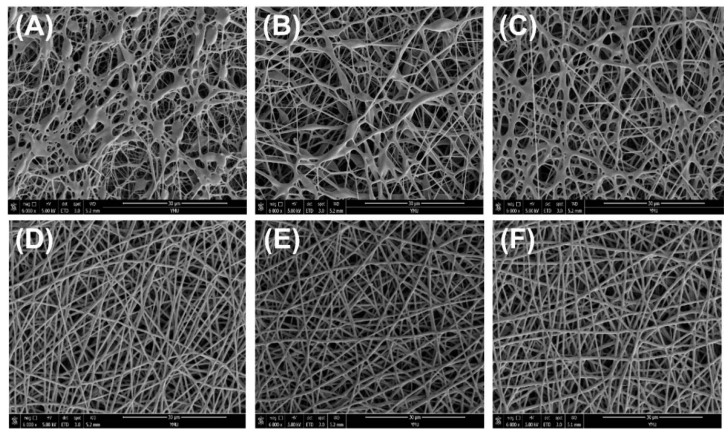
SEM image of electrospun fibre membrane. (**A**) PLLA, (**B**) PLLA/PTMC (9:1), (**C**) PLLA/PTMC (8:2), (**D**) PLLA/PTMC (7:3), (**E**) PLLA/PTMC (6:4) and (**F**) PLLA/PTMC (5:5) (scale bar: 30 µm).

**Figure 2 polymers-14-04406-f002:**
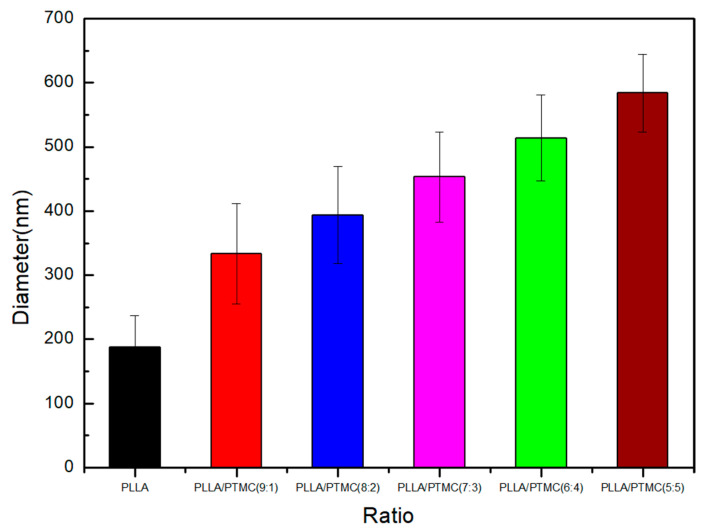
Fibre diameter distribution map. *p* < 0.05 with respect to PLLA, PLLA/PTMC (9:1) and PLLA/PTMC (8:2); *p* < 0.01 with respect to PLLA/PTMC (7:3), PLLA/PTMC (6:4) and PLLA/PTMC (5:5); error bars indicate SD; n = 5.

**Figure 3 polymers-14-04406-f003:**
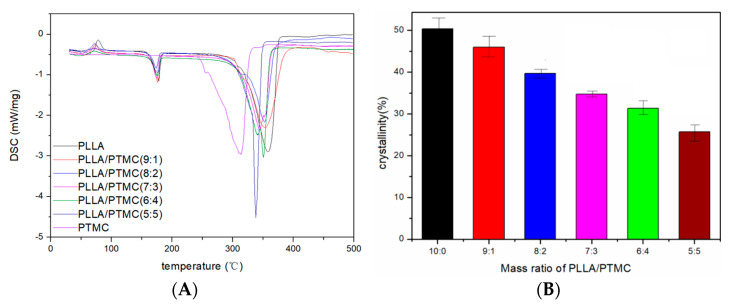
Differential scanning calorimetry curves (**A**) and crystallinity (**B**) of PLLA/PTMC electrospun fibres. *p* < 0.05; error bars indicate SD; n = 3.

**Figure 4 polymers-14-04406-f004:**
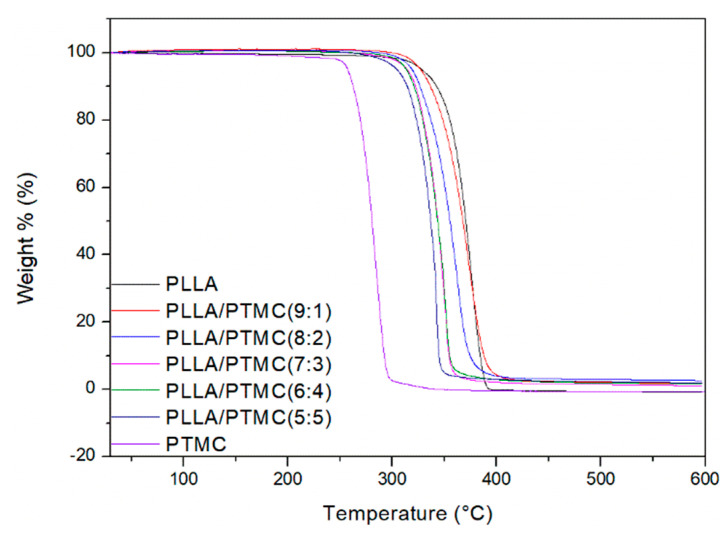
Results of thermogravimetric analyses conducted on PLLA/PTMC blend membranes characterised by different blending ratios. PLLA: poly(L-lactide); PTMC: poly(trimethylene carbonate).

**Figure 5 polymers-14-04406-f005:**
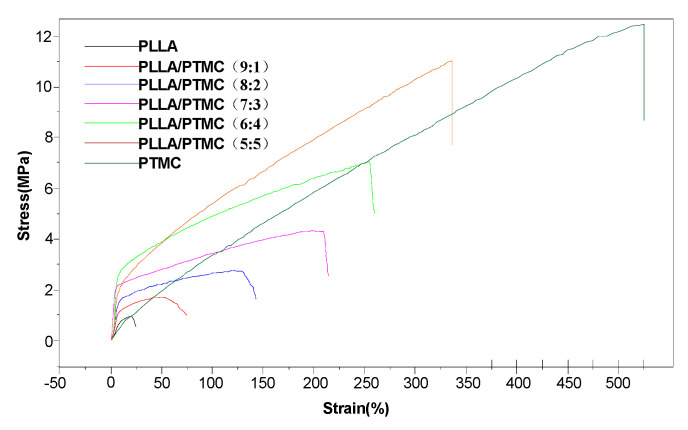
Stress–strain curves of PLLA/PTMC electrospun fibre membrane scaffolds.

**Figure 6 polymers-14-04406-f006:**
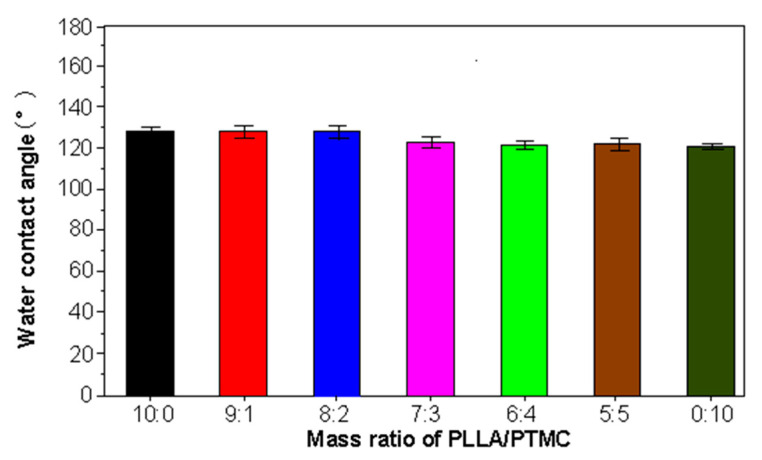
Measured water contact angle values for PLLA/PTMC blend membranes characterised by different blending ratios. *p* < 0.05; error bars indicate SD; n = 3.

**Figure 7 polymers-14-04406-f007:**
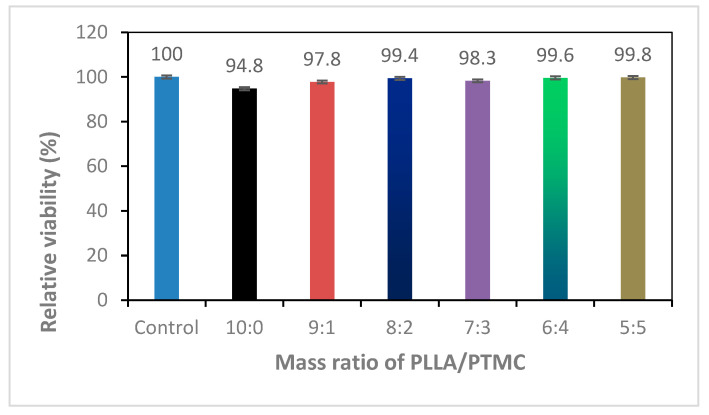
Bar diagram of the cytotoxicity tests conducted using PLLA/PTMC electrospun fibre membrane materials. *p* < 0.01 with respect to Control, 10:0, 8:2 and 7:3; *p* < 0.05 with respect to 9:1, 6:4 and 5:5; error bars indicate SD; n = 3.

**Figure 8 polymers-14-04406-f008:**
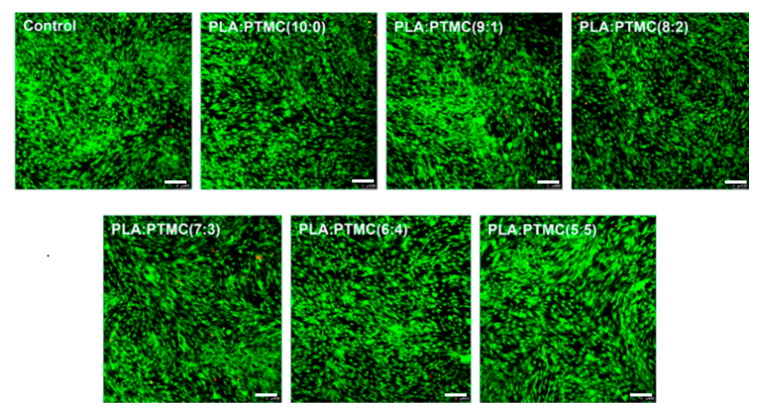
Photos taken with a laser confocal microscope of cells cultured on PLLA/PTMC fibre membrane scaffolds characterised by different blending ratios. (scale bar: 200 µm).

**Figure 9 polymers-14-04406-f009:**
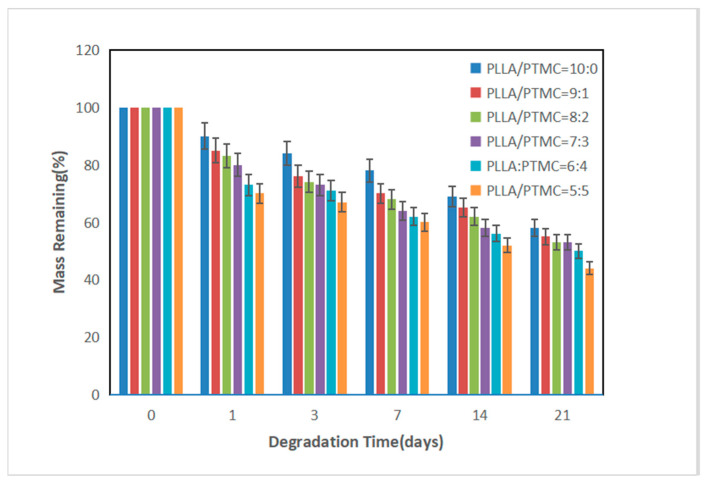
Mass remaining after degradation in PBS. *p* < 0.05; error bars indicate SD; n = 5.

**Table 1 polymers-14-04406-t001:** Mechanical properties of PLLAa/PTMCb electrospun fibre membrane scaffolds.

Scaffold Material	Modulus of Elasticity (MPa)	Tensile Stress (MPa)	Elongation at Break (%)
PLLA	10.13 ± 2.30	0.49 ± 0.07	24.71 ± 0.76
PLLA/PTMC (9:1)	26.37 ± 6.08	1.05 ± 0.26	72.69 ± 6.35
PLLA/PTMC (8:2)	35.28 ± 5.01	1.59 ± 0.13	137.51 ± 5.01
PLLA/PTMC (7:3)	65.17 ± 2.85	2.72 ± 0.24	202.55 ± 11.40
PLLA/PTMC (6:4)	74.49 ± 8.22	5.08 ± 0.44	257.88 ± 14.76
PLLA/PTMC (5:5)	31.31 ± 6.04	8.81 ± 1.40	344.85 ± 32.70
PTMC	5.07 ± 1.26	9.96 ± 1.54	552.54 ± 41.98

**Table 2 polymers-14-04406-t002:** Data reflecting the cytotoxicity of PLLA/PTMC blend membranes characterised by different blending ratios incubated for 72 h with MC3T3-E1 cells in vitro.

Group	OD ^a^	RGR ^b^ (%)	Toxicity Grade
Control	1.36900 ± 0.00557	100	-
PLLA	1.29767 ± 0.00907	94.78938	1
PLLA/PTMC (9:1)	1.36133 ± 0.04460	99.43998	1
PLLA/PTMC (8:2)	1.36100 ± 0.01039	99.41563	1
PLLA/PTMC (7:3)	1.34600 ± 0.00693	98.31994	1
PLLA/PTMC (6:4)	1.36300 ± 0.02193	99.56172	1
PLLA/PTMC (5:5)	1.36667 ± 0.02050	99.82956	1

^a^ Optical density, ^b^ Cell relative growth rate.

## Data Availability

The data presented in this study are available on request from the corresponding author.

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
