# Peer review of "Preparation and Properties of Electrospun PLLA/PTMC Scaffolds"

_polymers, 2022, doi:10.3390/polym14204406_

Round 1

Reviewer 1 Report

Authors reported the fabrication of Poly(L-lactide) (PLLA) and PLLA/poly(trimethylene carbonate) (PTMC) scaffolds were prepared by electrospinning system. The obtained results were confirmed by various techniques, such as SEM, TGA, DSC, and mechanical properties of the prepared scaffolds. I recommend the acceptance of the manuscript after major revision to address the following points: 

1. The abstract should contain some quantitative information. 

2. The novelty of the work should be described in the introduction. 

3. The author should provide the reference in the section 2.2. Preparation of PLLA/PTMC fibre membrane materials by electrospinning.

4.The author should provide the bar diagram of the Cytotoxicity tests conducted on PLLA/PTMC electrospun fibre membrane materials. 

5. The author should mention the scale bar in Figures 1 and 7.

6. Figures 2,3, and 4 resolution should be increase.

7. The Conclusions section should be fully revised with obtained results.

8. The author should provide the appropriate references in the following discussion sections. 3.1. Morphological characteristics of PLLA/PTMC electrospun fibre membrane material. 3.3. Mechanical uniaxial tensile tests conducted on PLLA/PTMC electrospun fibre membrane materials.3.4. Wettability experiments conducted on PLLA/PTMC electrospun fibre membrane materials. 3.5. Cytotoxicity of PLLA/PTMC electrospun fibre membrane materials.

9. It can be seen from Fig. 7 that on day 3 cell proliferation by LIVE/DEAD assay showed that cells have branched and interconnected network and good morphology suggesting increased proliferation rate.32 Electrospun PLLA/PTMC scaffolds showed good cytocompatibility in all blends ratio and cells were found well populated on PLLA/PTMC surface. Cells are sensitive to their surrounding while there was no significant morphological difference observed as compared to PLLA and control group in all the blends. The above sentences found the similarity from previously published work ("Fabrication of Robust Poly L-Lactic Acid/Cyclic Olefinic Copolymer (PLLA/COC) Blends: Study of Physical Properties, Structure, and Cytocompatibility for Bone Tissue Engineering", Journal of Materials Research and Technology, 2021). The author should re-write these sentences.

Author Response

June 14, 2022

Dear Sir/Madam,

Thank you for reviewing our manuscript and offering valuable advice. In accordance with your suggestions, we have made the following revisions to our manuscript:

  1. The abstract should contain some quantitative information.

Response: I have rewritten the abstract according to your request.

  1. The novelty of the work should be described in the introduction.

Response: The novelty of the work should be described in the introduction. “However, while studying the literature, we found that the related research on improving the tenacity of PLLA electrospinning fibers by adding PTMC to PLLA has not been reported yet.” We believe it is necessary to supplement this research node.

  1. The author should provide the reference in the section 2.2. Preparation of PLLA/PTMC fibre membrane materials by electrospinning.

Response:We have added a literature on PLLA/PCL electrospinning fiber preparation in Section 2.2. (Qi H, Ye Z, Ren H, Bioactivity assessment of PLLA/PCL/HAP electrospun nanofibrous scaffolds for bone tissue engineering, Life Sciences, 2016, 148, 139-144.)

4.The author should provide the bar diagram of the Cytotoxicity tests conducted on PLLA/PTMC electrospun fibre membrane materials.

Response: We supplemented the histogram of the cytotoxicity test of the electrospun fiber membrane material.(Figure 7)

  1. The author should mention the scale bar in Figures 1 and 7.

Response:We supplemented the scale bar information in Fig. 1 and Fig. 7.

  1. Figures 2,3, and 4 resolution should be increase.

Response: We have increased the clarity of Figures 2, 3, and 4.

  1. The Conclusions section should be fully revised with obtained results.

Response: We have rewritten the conclusion section.

  1. The author should provide the appropriate references in the following discussion sections. 3.1. Morphological characteristics of PLLA/PTMC electrospun fibre membrane material. 3.3. Mechanical uniaxial tensile tests conducted on PLLA/PTMC electrospun fibre membrane materials.3.4. Wettability experiments conducted on PLLA/PTMC electrospun fibre membrane materials. 3.5. Cytotoxicity of PLLA/PTMC electrospun fibre membrane materials.

Response: We have supplemented appropriate references as requested.

  1. It can be seen from Fig. 7 that on day 3 cell proliferation by LIVE/DEAD assay showed that cells have branched and interconnected network and good morphology suggesting increased proliferation rate.32 Electrospun PLLA/PTMC scaffolds showed good cytocompatibility in all blends ratio and cells were found well populated on PLLA/PTMC surface. Cells are sensitive to their surrounding while there was no significant morphological difference observed as compared to PLLA and control group in all the blends. The above sentences found the similarity from previously published work ("Fabrication of Robust Poly L-Lactic Acid/Cyclic Olefinic Copolymer (PLLA/COC) Blends: Study of Physical Properties, Structure, and Cytocompatibility for Bone Tissue Engineering", Journal of Materials Research and Technology, 2021). The author should re-write these sentence

Response:We have rewritten these sentence.

Thank you again for your valuable comments and suggestions.

Yours sincerely,

Dengbang Jiang

Reviewer 2 Report

Comments on polymers-1762469

In this manuscript, the authors prepared a series of electrospun PLLA/PTMC scaffolds with different PLLA: PTMC ratios. Several characterization methods were then used to evaluate the properties of these scaffolds. The manuscript was written logically, and the data were presented in a well-organized order. Several issues need to be addressed before publication.

Major points:

1.     The authors stated that “PTMC was self-made by the laboratory”. A detailed description of the synthesis and purification of PTMC should be added in the experimental section.

2.     The degradation mechanism and how long the material can be degraded in vivo should be discussed.

3.     A series of PTMC/PLLA fibers were fabricated. The authors need to discuss why the ratio of PTMC is only up to 50%, not higher.

4.     The authors need to discuss which formulation(s) have optimal properties for certain applications to show the advantages of blending PTMC and PLLA.

Minor points:

1.     Several characterization methods were mentioned in Section 2.3. It would be better to split it into several sections with one section for only one method.

2.     Line 109-112, 114-116, 123, 126-129: “the… was…” is one complete sentence and should be ended with a period, not a comma.

3.     Line 135, 137-140: Please complete these sentences. The experiment section is not supposed to be written with imperative sentences.

4.     Figure 1: Scale bars should be added to the image with values clearly labeled either in the image or in the figure caption. The current scale bar is not visible.

5.     No standard deviation was shown in Figure 3.

6.     Line 185-187: This sentence has grammar errors.

7.     No standard deviation was shown in Figure 6.

8.     Figure 7: Scale bars should be added to the image with values clearly labeled either in the image or in the figure caption. The current scale bar is not visible.

Author Response

June 14, 2022

Dear Sir/Madam,

Thank you for reviewing our manuscript and offering valuable advice. In accordance with your suggestions, we have made the following revisions to our manuscript:

Major points:

  1. The authors stated that “PTMC was self-made by the laboratory”. A detailed description of the synthesis and purification of PTMC should be added in the experimental section.

Response:We supplement the preparation and purification methods of PTMC.

  1. The degradation mechanism and how long the material can be degraded in vivo should be discussed.

Response:We supplemented the degradation experiments of the material.

  1. A series of PTMC/PLLA fibers were fabricated. The authors need to discuss why the ratio of PTMC is only up to 50%, not higher.

Response: Mainly due to the following two reasons,(1) Our purpose is to improve the brittleness of PLLA spinning fibers by adding PTMC, so PLLA as the base material accounts for more than 50%; (2) When the mechanical properties of the material were characterized, it was found that when the PTMC content reached 50%, the elastic modulus of the spinning fiber began to drop sharply, and the mechanical properties of the material deteriorated.

  1. The authors need to discuss which formulation(s) have optimal properties for certain applications to show the advantages of blending PTMC and PLLA.

Response: We discuss this in the conclusion section. The PLLA/PTMC electrospun fiber scaffold containing 40% PTMC has high elastic modulus and excellent toughness, and is expected to be applied in bone tissue engineering scaffolds and cell culture scaffolds.

Minor points:

  1. Several characterization methods were mentioned in Section 2.3. It would be better to split it into several sections with one section for only one method.

Response:We divide this part into four parts according to different characterization methods.

  1. Line 109-112, 114-116, 123, 126-129: “the… was…” is one complete sentence and should be ended with a period, not a comma.

Response:We have fixed these bugs.

  1. Line 135, 137-140: Please complete these sentences. The experiment section is not supposed to be written with imperative sentences.

Response:We rewrote these sentences.

  1. Figure 1: Scale bars should be added to the image with values clearly labeled either in the image or in the figure caption. The current scale bar is not visible.

Response:We have increased the resolution of Figure 1 and indicated the scale.

  1. No standard deviation was shown in Figure 3.

Response: We have supplemented the standard deviation in the figure.

  1. Line 185-187: This sentence has grammar errors.

Response:We have rewritten this sentence.

  1. No standard deviation was shown in Figure 6.

Response: We have supplemented the standard deviation in the figure.

  1. Figure 7: Scale bars should be added to the image with values clearly labeled either in the image or in the figure caption. The current scale bar is not visible.

Response:We have supplemented the scale bar in the figure.

Thank you again for your valuable comments and suggestions.

Yours sincerely,

Dengbang Jiang

Round 2

Reviewer 2 Report

In this revised manuscript, the authors have addressed all my previous comments. Therefore, I recommend acceptance of this revised manuscript.

Author Response

Dear reviewer:
Thank you for your decision and constructive comments on my manuscript.
We apologize for the poor language of our manuscript. We worked on the manuscript for a long time and the repeated addition and removal of sentences and sections obviously led to poor readability. We have now worked on both language and readability and have also involved native English speakers for language corrections. We really hope that the flow and language level have been substantially improved.
Thanks again! 
